# The information theory of developmental pruning: Optimizing global network architectures using local synaptic rules

**Carolin Scholl**[1], **Michael E. Rule**[2], **Matthias H. Hennig**[3]*

**1** Max Planck School of Cognition, Leipzig, Germany, **2** University of Cambridge, Engineering Department, Cambridge, United Kingdom, **3** University of Edinburgh, Institute for Adaptive and Neural Computation, Edinburgh, United Kingdom

* m.hennig@ed.ac.uk

**Data Availability Statement:** All code required to reproduce the results presented in the paper is available at https://github.com/carolinscholl/PruningBMs.

## Abstract

During development, biological neural networks produce more synapses and neurons than needed. Many of these synapses and neurons are later removed in a process known as neural pruning. Why networks should initially be over-populated, and the processes that determine which synapses and neurons are ultimately pruned, remains unclear. We study the mechanisms and significance of neural pruning in model neural networks. In a deep Boltzmann machine model of sensory encoding, we find that (1) synaptic pruning is necessary to learn efficient network architectures that retain computationally-relevant connections, (2) pruning by synaptic weight alone does not optimize network size and (3) pruning based on a locally-available measure of importance based on Fisher information allows the network to identify structurally important vs. unimportant connections and neurons. This locally-available measure of importance has a biological interpretation in terms of the correlations between presynaptic and postsynaptic neurons, and implies an efficient activity-driven pruning rule. Overall, we show how local activity-dependent synaptic pruning can solve the global problem of optimizing a network architecture. We relate these findings to biology as follows: (I) Synaptic over-production is necessary for activity-dependent connectivity optimization. (II) In networks that have more neurons than needed, cells compete for activity, and only the most important and selective neurons are retained. (III) Cells may also be pruned due to a loss of synapses on their axons. This occurs when the information they convey is not relevant to the target population.

## Author summary

Biological neural networks need to be efficient and compact, as synapses and neurons require space to store and energy to operate and maintain. This favors an optimized network topology that minimizes redundant neurons and connections. Large numbers of extra neurons and synapses are produced during development, and later removed as the brain matures. A key question to understand this process is how neurons determine which synapses are important. We used statistical models of neural networks to simulate

**Funding:** This work was supported the Engineering and Physical Sciences Research Council grant EP/L027208/1 (MHH). CS was supported by Studienstiftung des Deutschen Volkes (https://www.studienstiftung.de), BMBF (https://www.bmbf.de) and the Max Planck Society (https://www.mpg.de). The funders had no role in study design, data collection and analysis, decision to publish, or preparation of the manuscript.

**Competing interests:** The authors have declared that no competing interests exist.

developmental pruning. We show that neurons in such networks can use locally available information to measure the importance of their synapses in a biologically plausible way. We demonstrate that this pruning rule, which is motivated by information theoretic considerations, retains network topologies that can efficiently encode sensory inputs. In contrast, pruning at random, or based on synaptic weights alone, was less able to identify redundant neurons.

## Introduction

The number of neurons and synapses initially formed during brain development far exceeds those in the mature brain [1]. Up to half of the cells and connections are lost due to pruning [2–7]. This process of initial over-growth and subsequent reduction suggests that the optimal wiring of the brain is not entirely predetermined. Instead, experience-dependent plasticity allows for a topological refinement of neural circuits, thereby adapting them to the animal's environment [8]. The removal of unneeded neurons and synapses reduces the high costs of the brain in terms of material and metabolism [9]. Pruning may also amplify neuronal signals against synaptic noise and support competition among synapses, improving input selectivity [10]. Pathological over-pruning, however, may result in neuronal dysfunction, manifesting as cognitive impairments and clinical disorders such as schizophrenia [8, 11–13]. There seems to be a sweet spot between pruning and over-pruning for the brain to remain adaptive and resilient against damage.

What characterizes the cells and synapses that survive as opposed to the ones that die? A key factor for survival is neuronal activity. Initially, spontaneous activity is thought to drive survival. The refinement of cortical circuits then relies increasingly on sensory-driven and thus experience-dependent neuronal activity [14]. Neurons are thought to compete to activate postsynaptic targets in order to receive trophic supply [15–17]. Although highly active presynaptic neurons are favored in this competition, the strengthening of connections also depends on coincident activity of the postsynaptic neuron. For instance, when postsynaptic cells in the primary visual cortex were pharmacologically inhibited, their more active afferents were weakened [18]. Activity-dependent plasticity is thus based on bidirectional signaling between the presynaptic neuron and the postsynaptic neuron.

Here, we explore several local and biologically plausible rules for iteratively pruning unimportant synapses and units from artificial neural network models. Many models of neural pruning simply remove small synaptic weights [19–22]. However, weight magnitude is not the same as importance: small weights can be necessary to maintain accuracy [23]. A more theoretically grounded measure of synapse importance may lie in the activity of individual neurons and their correlations [24].

Information-theoretic approaches to network reduction provide a principled starting point. For example, the Optimal Brain Surgeon algorithm [25] estimates each parameter's importance by perturbing its value and re-evaluating an error function: low changes in error indicate redundant, uninformative parameters. This curvature with respect to small parameter changes is given by the Hessian of the error function, and is locally approximated by its negative, the Fisher Information Matrix (FIM). To reduce computational complexity, "Optimal Brain Damage" (as opposed to Optimal Brain Surgeon [23, 26, 27]) makes the simplifying assumption of a diagonal Hessian matrix.

Estimates of parameter importance based on Fisher Information (FI) have recently been used to overcome catastrophic forgetting in artificial neural networks [28]. In contrast to

identifying parameters worth keeping, we aim to identify parameters worth removing. In this work, we employ Restricted Boltzmann Machines (RBMs) [29] as an artificial neural network model to study activity-dependent pruning. For RBMs, the diagonal of the FIM can be estimated locally based on the firing rates of units and their coincidence [30, 31]. In principle, this estimate of parameter importance is available to individual synapses and neurons, and could thus guide the search for efficient representations during neurodevelopmental pruning.

Similar to neuronal sensory systems, RBMs extract and encode the latent statistical causes of their inputs. They consist of two layers of stochastic binary units, resembling the spiking of neurons. The visible units correspond to sensory input, while the hidden units encode a latent representation of this data. By adjusting their hidden representations to maximize the likelihood of the data [32], RBMs learn in an unsupervised fashion that resembles the Hebbian learning seen in neural networks [33, 34]. RBMs have been used as statistical models of sensory coding [30, 35–37]. Furthermore, multiple RBMs can be stacked to build a deep Boltzmann machine (DBM) [38] to model the hierarchical, bidirectional computation of the visual system [39, 40]. Such models have been used, e.g., to explore consolidation processes in declarative memory [41] or how loss of input could lead to visual hallucinations in Charles Bonnet Syndrome [42].

We organize the results as follows: we first introduce our estimates of synaptic importance based on locally-available activity statistics. We then discuss the overall network curvature and demonstrate that important synapses center on overall highly informative neurons. Based on these observations, we introduce local pruning rules to iteratively remove synapses and neurons from RBMs and DBMs that were trained on image patches from two different data sets. We evaluate the fit of the pruned models across different pruning criteria by assessing their generative and encoding performance. Finally, we discuss the biological plausibility of our activity-dependent pruning rules by comparing them to alternative rules used in our own and related work and provide implications of our findings.

## Results

### An activity-based estimate of Fisher information

The goal of this work is to derive and use a local pruning rule to reduce network size and identify a relevant network topology in restricted and deep Boltzmann machines. By 'relevant network topology' we mean a topology optimized for computational needs that includes only neurons and synapses that are relevant for the task at hand. In our experiments this task is the encoding of visual stimuli in hidden representations of RBMs and DBMs. RBMs are energy-based models whose energy function is given by:

$$E_{\mathbf{v},\mathbf{h}}^{\phi} = -\sum_{i=1}^{n_v} b_i^v v_i - \sum_{j=1}^{n_h} b_j^h h_j - \sum_{i=1}^{n_v}\sum_{j=1}^{n_h} w_{ij} v_i h_j, \tag{1}$$

where $v_i$ stands for visible unit $i$, $h_j$ stands for hidden unit $j$ and $w_{ij}$ for the bidirectional weight connecting them. Lower energy corresponds to higher probability of the respective model state. We work with Bernoulli RBMs, and all neurons are binary: $v_i$ and $h_j$ are either firing (1) or silent (0). The biases $b_i^v$ and $b_j^h$ correspond to the excitability of a neuron. All weights and biases make up the parameter set $\phi = \{\mathbf{W}, \mathbf{b^v}, \mathbf{b^h}\}$.

The Hessian of the objective function of a model gives information about the importance of parameters, and can be used for pruning [23, 25]. It can be locally approximated by its negative, the FIM. It was recently shown that local activity statistics indeed correlate with importance as assessed by Fisher information [30]. In the case of the RBM this correspondence is

exact; an entry of the FIM for an RBM has the form:

$$\mathbf{F}_{ij}(\phi) = \sum_{\mathbf{v},\mathbf{h}} P_{\mathbf{v},\mathbf{h}} \frac{\partial^2 E_{\mathbf{v},\mathbf{h}}}{\partial \phi_i \, \partial \phi_j}, \tag{2}$$

where $P_{\mathbf{v},\mathbf{h}}$ is the probability of jointly observing visible pattern $\mathbf{v}$ and hidden state $\mathbf{h}$. The FIM locally approximates the Kullback-Leibler divergence between the model distribution under the current parameter set $\phi$, and the model distribution when a pair of parameters $\phi_i$ and $\phi_j$ is slightly perturbed.

A crucial advantage of the FIM is that its values are closely connected to the distribution of activity and can be estimated by sampling from the distribution $P_{\mathbf{v},\mathbf{h}}$. In the case of the RBM, the Fisher information for two weights $w_{ij}$ and $w_{kl}$ takes the form:

$$\mathbf{F}_{w_{ij},w_{kl}} = \langle v_i h_j v_k h_l \rangle - \langle v_i h_j \rangle \langle v_k h_l \rangle. \tag{3}$$

For Fisher information to be a suitable indicator for biological neurons, it must also be available locally at each synapse. This precludes using the full FIM, which requires information about the connectivity of the whole network. However, the diagonal of the FIM is locally available. The diagonal entries of the FIM correspond to Eq (3), when the two modified parameter values are the same ($w_{ij} = w_{kl}$). It captures the effect on the error when we modify a single weight. For the FI of the weights, Eq (3) simplifies to [30]:

$$\mathbf{F}_{w_{ij},w_{ij}} = \langle v_i^2 h_j^2 \rangle - \langle v_i h_j \rangle^2 = \langle v_i h_j \rangle (1 - \langle v_i h_j \rangle), \tag{4}$$

since $v_i^2 = v_i$ and $h_j^2 = h_j$ because neural activation is binary. Single synapses can therefore access the diagonal of the FIM from local activity statistics, and potentially use this to inform developmental decisions.

## Large encoding models have many poorly specified parameters

In a first line of experiments, we inspected the curvature of the energy landscape of overcomplete models that had more latent units than needed to encode the visible stimuli. We started by fitting overcomplete RBMs to circular, binarized patches of images of natural scenes (see Fig 1A) using the wake-sleep algorithm [43]. For these relatively small models, we computed the full FIM, which typically turned out to be sparse (see Fig 1B). This indicates that large encoding models indeed have many poorly specified parameters, which can be safely removed from the model.

The sparseness of Fisher information is also evident from a visualization of the first eigenvector and the diagonal of the FIM. This is plotted separately for the weights, hidden biases and visible biases in Fig 1C and 1D. Furthermore, the eigendecomposition of the FIM revealed that the largest eigenvalue ($\lambda_0$) is larger than the second largest eigenvalue by an order of magnitude (first eigenvalue 23.97, second eigenvalue 1.60). This low-rank structure implies that the FIM can be approximated by a rank-1 outer product of its leading eigenvector, $\mathbf{u}$. This, in turn, implies that the diagonal entries (Eq 4) of the FIM correlate closely with parameter importance. This supports employing the diagonal as a proxy for parameter importance:

$$\mathbf{F} \approx \lambda_0 \mathbf{u}\mathbf{u}^\top \Rightarrow \mathrm{diag}[\mathbf{F}] \approx \mathrm{diag}[\lambda_0 \mathbf{u}\mathbf{u}^\top] \propto \mathbf{u}^2. \tag{5}$$

Strikingly, important synaptic weights were not uniformly distributed throughout the network. Instead, clusters of important weights (and biases) were associated with specific units (see Fig 1C and 1D). This suggests that a measure of synaptic importance based on Fisher

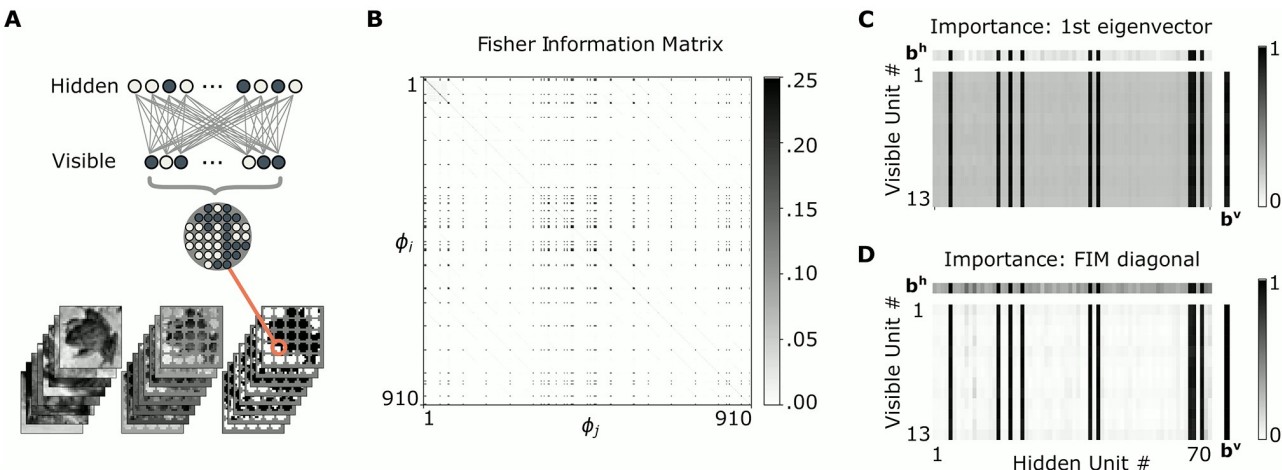

**Fig 1. Model architecture and visualization of Fisher information.** (A) Preparation of dataset and general model architecture: circles of different radii were sliced out of CIFAR-10 images and binarized. The pixels corresponded to the visible units of a standard RBM with one hidden layer. (B) Fisher information for an exemplary over-parameterized RBM initialized with $n_v = 13$ visible units and $n_h = 70$ hidden units. The FIM is sparse, indicating many irrelevant parameters. (C) Importance of each parameter, as summarized by the value for each parameter in the vector of steepest curvature in the FIM (i.e. the leading eigenvector). The rectangular plot shows the normalized importance for all weights $w_{ij}$ connecting visible units ($v_i$; vertical axis) and hidden units ($h_j$; horizontal axis). The importance for biases for the hidden ($\mathbf{b^h}$) and visible ($\mathbf{b^v}$) units is shown above (horizontal) and to the right (vertical), respectively. (D) Normalized parameter importance directly estimated from the diagonal entries of the FIM. (FIM = Fisher Information Matrix).

information separates important and unimportant hidden units. It follows that pruning based on Fisher information likely leads to the disconnection of entire units, which would allow their removal from the network. This would correspond to neuronal apoptosis after excessive synaptic pruning.

Overall, these empirical investigations reveal that there are only few important units, and that important weights align with them. This confirms that parameter importance as revealed by the FIM contains meaningful information for pruning, and predicts that synaptic pruning will eventually lead to the removal of whole neurons that lack relevant afferent or efferent connections.

## Local pruning rules based on Fisher information

We now present our results of applying the local estimate of Fisher information as a criterion to prune RBMs. We compare this estimate to removing poorly specified, unimportant weights according to different pruning criteria and random pruning. We also test a pruning criterion based on the full FIM to confirm that the diagonal approximation closely tracks importance. For this, we assess importance as the magnitude assigned to each weight in the leading eigenvector of the FIM.

Eq (4) gives weight importance based on the covariance between presynaptic and postsynaptic firing, $\langle v_i h_j \rangle$. We refer to this as the "variance" estimate of the FIM diagonal. However, neurons need not track this coincident firing explicitly. Activity-driven correlations influence synaptic weights and vice versa. As a result, the statistical quantities relevant to pruning are tied to the synaptic weights themselves. This opens up a broader range of possible biological implementations of the rules we explore here. It also highlights that apparent correlations between synaptic weights and pruning *in vivo* could arise from a FI-based pruning strategy. In Materials and Methods: *Heuristic estimate of Fisher Information* we derive a mean-field

approximation of the pre-post correlations that depends only on the synaptic weight and the average firing rates of the presynaptic and postsynaptic neurons:

$$\langle v_i h_j \rangle \approx \frac{\langle v_i \rangle \langle h_j \rangle}{\langle v_i \rangle + (1 - \langle v_i \rangle)\, e^{-w_{ij}(1 - \langle h_j \rangle)}}. \tag{6}$$

We show that this heuristic remains surprisingly accurate, offering a proof of principle that locally available quantities can provide the signals needed for a synapse (and by extension a neuron) to assess its own importance in the network. We refer to this alternative FI-based pruning rule as the heuristic FI estimate.

For our synaptic pruning experiments, we ranked weights according to their importance assessed by one of the aforementioned local estimates, or by the first eigenvector of the FIM. The unpruned full model initially had $n_v = 13$ visible units, $n_h = 70$ hidden units, and $n_w = n_v \times n_h = 910$ weights. We then iteratively deleted half of the weights with lowest estimated importance, while monitoring model fit.

We assessed the fit of a given model based on its generative performance, i.e. how well the generated patterns in the visible layer match the patterns available in the training data. Following the approach by Reichert et al. [42], we use Boltzmann machines as hierarchical generative models that may capture aspects of generative processing in the brain. In these models, good generative performance is equivalent to Shannon-optimal encoding [43]. The psychological interpretations of this are more speculative [42], and we discuss this later in the text.

Unconnected hidden units were removed from the model. Absolute weight magnitude, random weight pruning, and removal of the *most* important weights ('Anti-FI' pruning) served as reference criteria for pruning. We also added a control case of neuron pruning: instead of pruning random connections, whole units were randomly taken out of the network. Fig 2B and 2C show the model's average generative performance after three pruning iterations. Each simulation was repeated ten times.

Excessive pruning degraded generative performance. Regardless of the pruning criterion, the distributions of generated patterns vs. the training data generally diverged immediately after pruning (Fig 2C). However, subsequent retraining rescued models that were pruned according to our Fisher information estimates or weight magnitude: the Kullback-Leibler divergence between the data distribution and the distribution over visible samples eventually was comparable to the one of the initial model before pruning. In biological neural networks, training occurs continuously alongside activity-driven pruning, so a period of retraining following pruning is plausible.

For "random weight" and "Anti-FI" pruning, the distributions of generated patterns diverged from the training data and could only be partially rescued by retraining (see Fig 2C). This is reflected in the errors of the matched probabilities shown in Fig 2B. Each generated pattern was matched with the probability of the same pattern occurring in the training data. Generally the error was larger for rare patterns than for frequent ones. Agreement between the model and training data probabilities was particularly poor for "random weight" and "Anti-FI" pruning. Even some frequent patterns were no longer matched by wrongly pruned models. On the other hand, models that had small weights or uninformative weights removed reached a generative performance level comparable to the initial model shown in Fig 2A.

We conjectured that the FI-based rules allowed local synaptic dynamics to identify and remove unimportant neurons. To confirm this, we ran a simulation which removed a random subset of neurons. This showed a large divergence between the distribution of training patterns and generated patterns immediately after pruning, but performance could be restored by retraining (Fig 2C). This confirms that retaining a few, fully connected units is better than many,

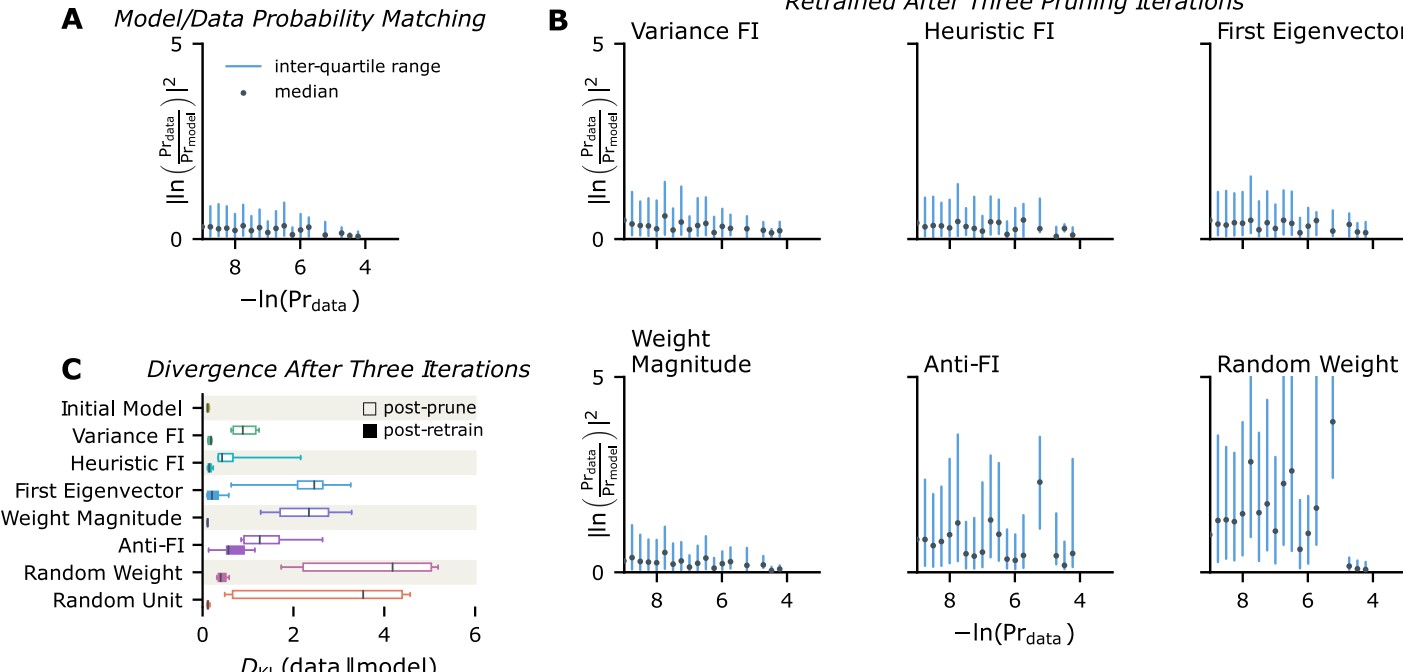

**Fig 2. Generative performance of networks before and after iterative pruning.** Half of the weights were removed in each of 3 pruning iterations; all plots reflect 10 independent simulations. (A) Mean-Squared-Error (MSE; vertical axis) between the log-probability of a pattern in the training data (horizontal axis) and the log-probability of the corresponding pattern generated by the trained model. Black points reflect the average MSE over 10 realizations; Blue lines indicate the 25th-75th percentile range. (B) Log-probability matching after three iterations of synaptic pruning (according to different criteria) and retraining. Accuracy remains good for pruning rules based on weight magnitude and Fisher information. (C) Kullback-Leibler divergence between the distribution over training instances vs. the distribution over generated samples, after three iterations of pruning before and after-retraining. Retraining generally restores generative performance.

sparsely connected ones in this model. This hints that the role of FI-pruning in densely-connected networks may be to select an optimal network size, rather than a specific sparse connectivity. Synaptic pruning rules that leave partially connected neurons have poorer performance.

While the generative performance after pruning and retraining was comparable for models pruned according to weight magnitude and Fisher Information, the resulting model architectures differed markedly in their number of hidden units. Table 1 shows the number of remaining hidden units $n_h$ after each of the three pruning iterations, averaged across ten simulations.

**Table 1. Differences in numbers of remaining hidden units.** Each table entry contains the mean "$\mu$" number of hidden neurons $n_h$ after each pruning iteration, and their standard deviation ($\sigma$) in parenthesis. The results were averaged over 10 simulations. The model initially had $n_v = 13$, $n_h = 70$, and $n_w = n_v \times n_h = 910$ weights. During each of the three pruning iterations half of the weights were removed according to different criteria. $n_w$ describes the remaining weights at the given phase of pruning.

| Pruning criterion | $n_w = 455$ | $n_w = 227$ | $n_w = 113$ |
|---|---|---|---|
| Variance FI | $\mu = 66.8$ ($\sigma = 4.14$) | 47.6 (5.75) | 10 (1.26) |
| Heuristic FI | 58.0 (4.12) | 42.3 (4.8) | 14.3 (7.06) |
| First eigenvector | 69.8 (0.4) | 42.5 (4.8) | 9.5 (0.67) |
| Weight magnitude | 69.9 (0.3) | 58.1 (2.7) | 32.2 (3.06) |
| Anti-FI | 55 (3.35) | 44 (2.69) | 33.6 (2.24) |
| Random Weight | 70 (0.0) | 69 (0.75) | 58.0 (2.79) |
| Random Unit | 35 (0.0) | 18 (0.0) | 9 (0.0) |

Although the number of remaining weights $n_w$ was matched across pruning criteria, different numbers of units were disconnected through repeated weight pruning. Fisher information typically concentrated on few hidden units (see Fig 1C and 1D), contrasting them with units of low overall importance. Such uninformative units were left disconnected after repeated pruning. In contrast, random weight pruning disconnected relatively few units. Pruning the largest weights (weight magnitude) and most important weights (Anti-FI) also retained more hidden units in the model, making the final encoding less efficient compared to FI-motivated pruning.

In sum, for all pruning strategies the network could recover to some extent from the loss of weights and units through retraining [44, 45]. However, the resulting latent representations differed: the number of disconnected neurons varies between the pruning criteria. This may affect subsequent read-outs from these representations, an effect we will investigate next using multi-layer networks.

## FI-guided pruning retains a useful encoding in multi-layer networks

Above, we demonstrated an iterative approach to pruning weights from RBMs while monitoring their fit from the frequency of its generated patterns.

In small RBMs that encoded simple patterns, retraining compensated for the loss of synapses regardless of the pruning criterion. However, we found that FI pruning, as opposed to pruning small weights, reduced the cost of the network in terms of the number of remaining neurons. This supports the view that the function of pruning is to find an appropriate architecture, rather than specific parameter configurations.

To investigate the effect of the pruning method on learned representations, we increased the complexity of the model architecture by adding another hidden layer, resulting in a deep Boltzmann machine. We trained this multi-layer model on a labeled dataset to quantify the fit of the model and the quality of the latent representation during iterative pruning. Specifically, we used the MNIST handwritten digits dataset [46], and tested both the generative and classification performance using the latent activations. To measure encoding quality, we compared the accuracy of a logistic classifier trained on the stimulus-evoked activity in the latent units ("encodings") to one trained on the raw digits.

Each image of the dataset was binarized and scaled to 20×20 pixels. To simplify computation and add biological realism, we restricted the receptive fields of each unit in the first hidden layer $\mathbf{h}^1$ to a small region of the visible inputs. To encode digits, the network was therefore forced to combine these lower-level features in the second hidden layer $\mathbf{h}^2$ (see Fig 3A). We then assessed encoding quality based on the accuracy of a classifier applied to the stimulus-driven activity in the deepest hidden layer. Before pruning, the classifier performed better using the latent states of the trained network compared to the raw digits. Fig 3B shows the classification errors over the course of pruning according to different criteria. After each pruning event the models were retrained for 10 epochs and evaluated.

On each iteration, we removed the least important 10% of weights as assessed by weight-specific FI or absolute weight magnitude in $\mathbf{h}^1$, and the least important 25% of weights in $\mathbf{h}^2$. For the FI-based rules, more than 25% of FI estimates of weights in $\mathbf{h}^2$ were zero in the first pruning iteration. In this scenario, we instead pruned all synapses with zero importance, which led to a comparably faster rate of model reduction. Units of the intermediate layer $\mathbf{h}^1$ that became disconnected from either side ($\mathbf{v}$ or $\mathbf{h}^2$), were completely removed. Thus, the total number of deleted weights may be larger than specified by the importance measure as a

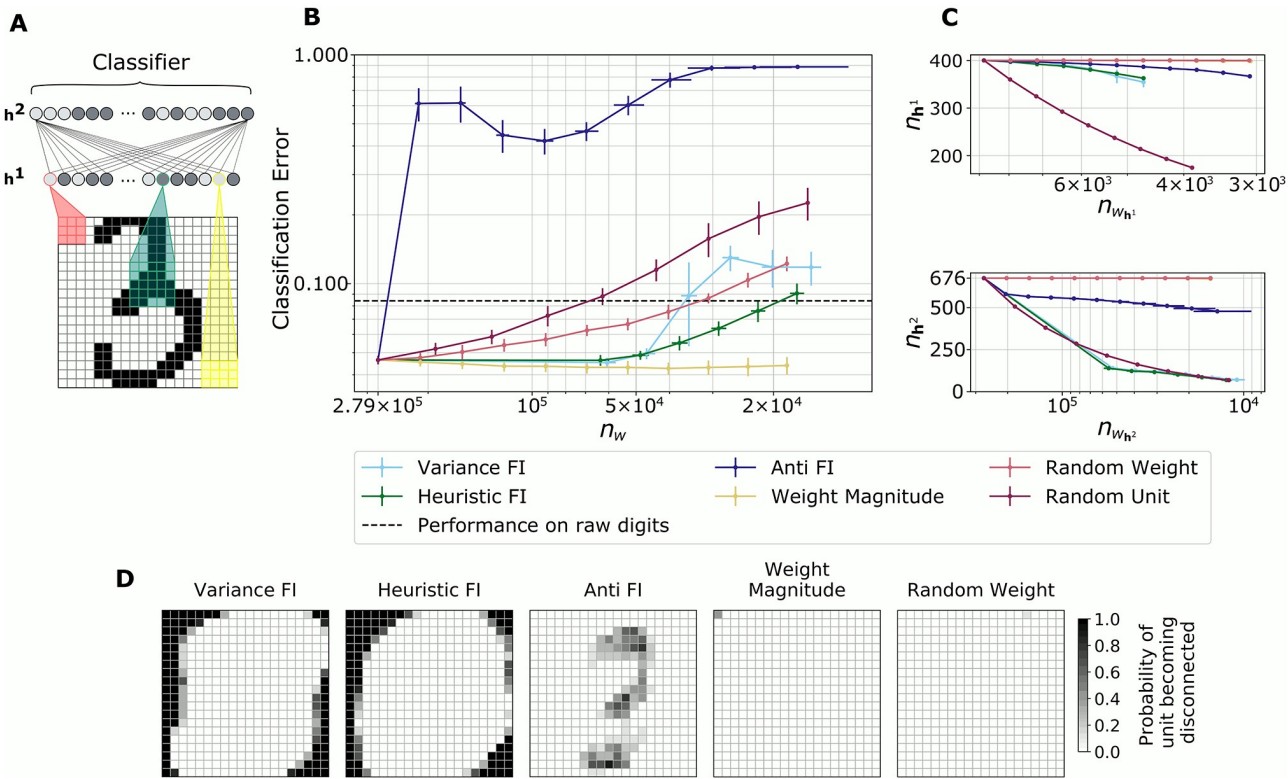

**Fig 3. Encoding performance during pruning.** (A) Data representation and deep Boltzmann machine architecture. MNIST digits were cropped and binarized. Each unit from hidden layer $\mathbf{h^1}$ had an individual receptive field covering neighboring input pixels; $\mathbf{h^2}$ was fully connected. Only a few connections are shown due to clarity. The classifier was trained on latent encodings from $\mathbf{h^2}$. (B) Classification error of a logistic regression classifier trained on $\mathbf{h^2}$ encodings as a function of remaining weights $n_w$. The dotted line stands for the baseline error of a classifier trained on the raw digits. All data points are averages from 10 independent simulations, and error bars denote one standard deviation. (C) Number of latent units in $\mathbf{h^1}$ and $\mathbf{h^2}$ as a function of remaining weights over the course of pruning. (D) Final visible layer connectivity after pruning according to different criteria. The probability of a unit being disconnected is shown in gray-scale, with black denoting units that were disconnected in all simulations.

secondary effect. The removal of such "dead-end" units is also the reason for the high variability of the number of remaining weights for the Anti FI pruned models (see Fig 3B).

Similarly to the results for the single-layer RBM, pruning a modest number of parameters had little effect on the latent encoding, as measured by the classification error (Fig 3B). Performance usually decreased immediately after pruning (not shown), but was restored by retraining. With successive pruning iterations, performance decreased steadily. Interestingly, pruning according to absolute weight magnitude retained a useful encoding for the classifier; its error even slightly decreased in the course of ten pruning iterations. Yet very few units were disconnected and pruning to a comparable size of remaining weights required more iterations than pruning according to our FI estimates.

In contrast, random removal of weights or complete units led to rapid degradation, and pruning important weights (Anti-FI pruning) was particularly harmful: classification performance deteriorated after the first pruning event, and eventually remained below chance even with retraining. This suggests that Anti-FI pruning causes a loss of topologically relevant neurons. The visualization of the pixels corresponding to the remaining units in the final visible layers $\mathbf{v}$ after pruning in Fig 3D shows this clearly: While FI-based pruning correctly

disconnects pixels at the boundaries where images contain no information, Anti-FI pruning removes links to the central pixels most useful for digit classification.

Fig 3C shows the number of remaining units as a function of remaining weights in the two latent layers of the DBMs. The difference in the number of lost units agrees with with what we observed for RBMs (Table 1). FI-guided pruning allowed for the removal of units in all layers of the model, making the encoding more efficient.

The encoding also remained useful for classification. When weights were pruned according to our heuristic estimate of Fisher information, the performance of the classifier only fell below baseline after the number of weights was reduced by more than one order of magnitude. Models pruned according to the variance FI estimate dipped in their performance after the third iteration of pruning. Arguably, the optimal model size was already reached after the first pruning iteration. We hypothesize that an increase of the average activation of hidden units may indicate the arrival at the optimal model size and thus be a signal to stop pruning (see Fig A in S1 Text). Activation of hidden units in $\mathbf{h}^1$ is less sparse from the beginning, as the receptive fields pre-structure the network and the number of hidden units in this layer equals the number of visible units (average activity in $\mathbf{h}^1$ in the first pruning iteration ranged between 0.37–0.43 as opposed to 0.0006–0.018 in $\mathbf{h}^2$). Initial activity in $\mathbf{h}^2$ is sparse and as a consequence FI pruning quickly identifies all units with low (covariant) activity: their weights are pruned and the unimportant units are removed from the model. This also allows a faster reduction of model size: to reach a comparable total number of weights, only six pruning iterations were necessary for our FI estimates, but ten iterations for the random weight and weight magnitude pruning.

FI pruning topologically optimizes the model: more weights (and units) are removed in $\mathbf{h}^2$ (see Fig 3C), while in $\mathbf{h}^1$ the pruning rate agrees with the pre-specified weight reduction per iteration (10%). Such a topological optimization within multiple layers cannot be achieved by random unit removal. In this control case, where we pruned a comparable number of weights by randomly deleting hidden units, too many units were removed from $\mathbf{h}^1$ for the encoding to remain useful for classification.

Random weight pruning and pruning by absolute weight magnitude failed to reduce the model size in terms of neuron number: Neither pruning criterion led to any of the hidden units becoming disconnected. In the visible layer, a maximum of one pixel became disconnected in each case (see Fig 3D). Pruning according to our estimates of FI on the other hand left a number of hidden units as well as uninformative visible units at the borders of the images unconnected.

Taken together, these results show that the pruning criterion matters in deeper networks. The network performance can recover through retraining if activity-dependent pruning is used (FI and weight-magnitude pruning), but is permanently damaged by random or Anti-FI pruning. Furthermore, FI-pruning produces the most efficient network topology and selectively retains the most important neurons, unlike simpler strategies such as pruning by weight magnitude.

## Models that lost informative synapses and neurons can no longer generate meaningful patterns

Previous studies explored deep Boltzmann machines as a generative model of hallucinations, for example those seen in Charles Bonnet Syndrome [42]. In this disease, patients experience vivid hallucinations after going blind. DBMs may be a suitable model for this because they are probabilistic generative models: without supervision, they learn a statistical representation in the hidden layers that can be used to generate the visible samples. To simulate the loss of

vision, Reichert et al. [42] removed the visible layer of a deep Boltzmann machine. Due to homeostatic compensation, spontaneous activity in the hidden units continued to reflect plausible visual inputs, which would be interpreted as hallucinations by downstream readouts.

Inspired by these results, we asked whether pruned networks retain a meaningful description of the latent structure in their inputs. We assessed this by measuring the generative performance of the networks over the course of pruning. To quantify generative performance, we asked a classifier trained on the raw digits to categorize samples and predict probabilities of them belonging to each digit class. These were summarized as digit-wise quality scores and diversity scores for the generated patterns (Materials and Methods: *Evaluation*).

We used the classification confidence as a proxy for digit quality (Fig 4A). When the model was deprived of its most informative weights (Anti-FI pruning), retraining could not compensate for the loss of relevant weights. A weaker, but significant degradation was also observed for random weight and random unit pruning. In contrast, the quality of generated digits by networks pruned by weight magnitude, or using the FI-based rules, suffered less after allowing for retraining. Arguably all networks were overpruned after pruning completed. Yet the generated patterns by the final models pruned according to our FI estimates or weight magnitude still partially resemble digits, as opposed to the ones produced by randomly pruned and anti-FI pruned models (see Fig 4C).

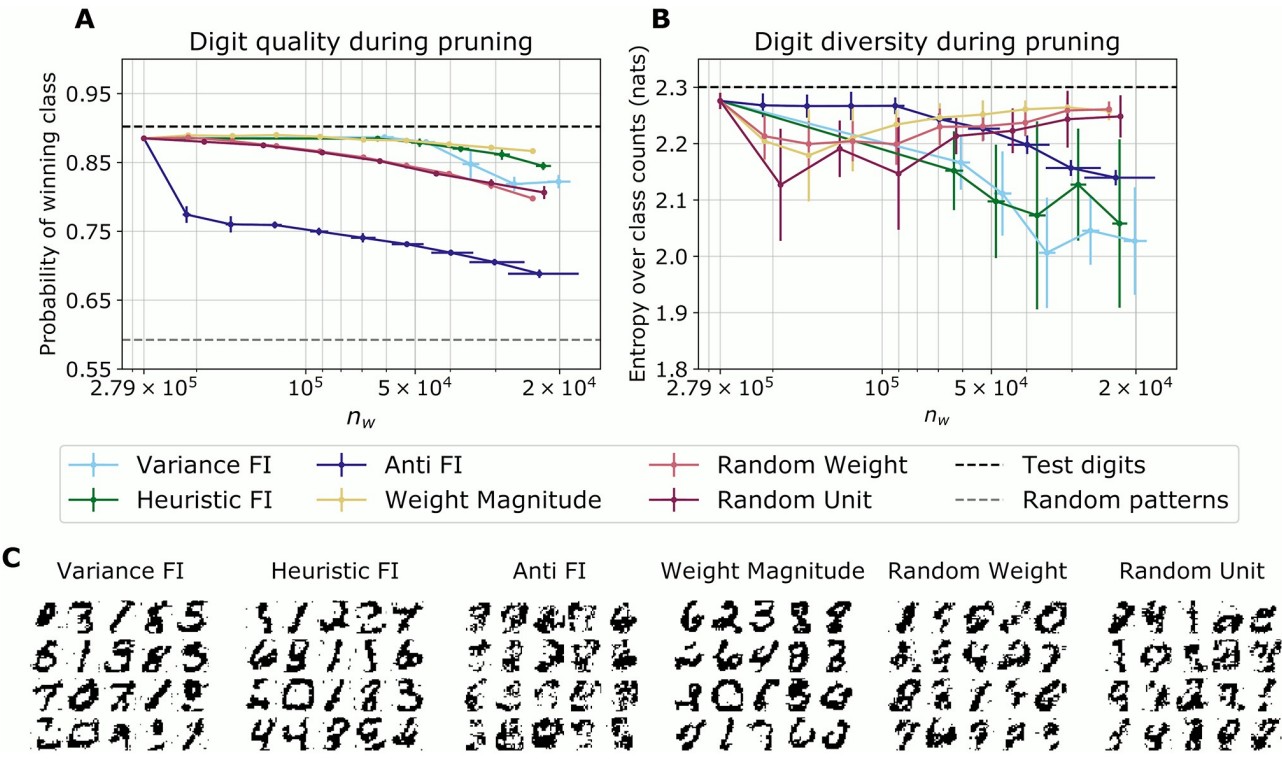

**Fig 4. Generative performance during pruning.** (A) Maximum class probability assigned to generated samples from pruned networks averaged over 10 runs. This summarizes the confidence of a classifier that the generated digits belonged to a specific class. Error bars denote standard deviations. The black and gray dashed line show the confidence of the classifier for the MNIST test digits and randomly generated patterns, respectively. (B) Entropy over the distribution of generated digits. An entropy value of $\approx 2.30$ nats corresponds to even coverage of all digits, which is achieved by the test digits (dotted line). All data points are averages from 10 independent simulations, and error bars denote one standard deviation. (C) Examples of generated patterns after pruning completed.

Finally, we compared the diversity of the generated digits (Fig 4B). We used a classifier to categorize each sample into one of the ten digits. In the unpruned model, the fraction of samples categorized as one of each of the ten digits ranged from from 7% to 18% across all runs, indicating good generative performance (if the samples were completely balanced, each digit would appear with a probability of 10%).

Pruning according to our FI estimates decreased digit diversity more so than the other criteria. Note, however, that this assessment of generative performance is not meaningful when the network generated degraded digits that were hard to classify, as was the case for Anti-FI or random pruning (compare examples in Fig 4C). For pruning based solely on weight magnitude however, both the quality and diversity of generated digits suffers less even after repeated pruning.

These results suggest that pruning impairs the generative capabilities of the networks, either degrading the generated representations (Fig 4A) or causing a bias towards certain patterns (Fig 4B).

Overall, FI-based pruning preserved the generative quality to a similar degree. In contrast, the Anti-FI pruning algorithm disconnected important units, leading to both sensory "blindness" and an inability of the model to meaningfully report the visual correlates of latent activity. Weight magnitude alone is an insufficient indication of parameter importance, and while networks pruned in this way still can be used to classify the digits well, they lack a structural optimization. In contrast, FI pruning resulted in a more efficient encoding in terms of the number of remaining units.

## Discussion

In this work, we used stochastic latent-variable models of sensory encoding to derive new theoretical results on neural pruning, and explored these results through numerical simulations. By examining the energy-based interpretation of such models, we showed that the importance of synapses and neurons can be computed from the statistics of local activity. To our knowledge, our work is the first empirical study of network reduction guided by an activity-dependent estimate of Fisher information in this context. Our pruning rule operates at single synapses, and provides a principled route to eliminating redundant units. Searching for biological analogues of these quantities and processes may provide insight into naturally occurring pruning in the developing brain.

### FI-guided pruning identifies efficient network architectures

In biology, over-production of neurons and synapses likely confers advantages. These extra neurons could accelerate initial learning [47], or allow the network to identify better architectures than it would otherwise [48]. After this initial learning, neurons and synapses not involved in covariant experience-dependent activity are lost [14, 18]. Our theoretical derivations make this precise, deriving local pruning rules that estimate the importance of a connection from the covariant activity of a pre- and postsynaptic neuron. This captures a central intuition about importance: neurons with low firing-rate variability are less relevant, and synapses with weak pre-post correlations are less important.

The importance of a precise pruning rule in identifying efficient network topologies is highlighted by the comparison of the different pruning criteria. Weight-based pruning yields networks with fewer synapses that still perform well, but unlike during FI-based pruning, few neurons are disconnected. Anti-FI pruning, in contrast, removes the most important synapses and vital units first, exactly those components that are left intact by FI-based learning rules. In our simulations, the irrelevant units are in the visual periphery as they carry no information.

However, parameter importance may be less obvious in deeper layers of the network, and more generally for other stimuli and tasks.

## Relationship to pruning in artificial neural networks

State-of-the-art models in machine learning are often over-parameterized, and optimizing such models via pruning is a subject of active research. The vast reduction of the number of neurons in the hidden layers in our FI-pruned models show how pruning could play a similar role in architecture optimization in the brain.

Yet some studies call into question the usefulness of pruning. It can be difficult to continue training networks with pruned weights without incurring performance losses [44, 45, 49], as networks become trapped in local minima. It has thus been argued that the value of pruning is rather to find a suitable architecture for random re-initialization [44, 45] or well-initialized weights of sub-networks within larger networks [49]. However, neither algorithm can be implemented by biological networks because they include a reset of synaptic weights to previous values. In our work, incremental retraining of trained parameter values was sufficient to preserve performance. This is consistent with the strategy that must be employed in the brain, in which learning and pruning occur concurrently. It remains to be seen whether the brain is also subject to the trap of local minima, or whether additional mechanisms make the issue irrelevant in biological networks.

## Biological plausibility of different pruning criteria

Pruning studies in artificial neural networks often estimate parameter importance with respect to the curvature of a cost function on network performance or discriminative power in supervised learning problems [23, 25, 50]. This requires access to a global state, and is not applicable to biological networks. In contrast, our study examined unsupervised learning in sensory encoding models that seek to model the latent causes of sensory input [32, 43, 51]. We showed that an energy-based perspective on such encoders leads naturally to an implicit measure of parameter importance, and that single synapses could compute this measure.

The physiological correlates of our FI-based pruning rule are unknown; to address this question, data from long-term imaging experiments of synapses could be fit to different potential pruning rules [52]. We find that activity-driven pruning leads to better structural optimization than simply removing small weights. Although parameter sensitivity correlates with weight values [31], we show that activity-dependent pruning that aims to identify uninformative neurons must consider more than the magnitude of a weight. Sampling the correlations between presynaptic and postsynaptic activity is one way to compute importance. A possible mechanism may be based on a correlation-sensitive calcium signal, which has been shown to have homeostatic effects when combined with structural plasticity [53]. We also show that importance can be estimated by combining weight magnitude with the firing rates of the presynaptic and postsynaptic neuron by our heuristic FI pruning criterion. This highlights that deceptively simple biological quantities can encode information-theoretic measures of synaptic importance.

We derived the FI-based pruning rules for Boltzmann machines, a class of energy-based stochastic neural network models that, owing to their symmetric connectivity, have a well defined equilibrium distribution and allow computing the FIM exactly. While the symmetric connectivity and learning rule (the wake-sleep algorithm, which requires symmetric connections) in these networks are not biologically plausible, they implement minimization of variational free energy as an unsupervised learning objective. This is a plausible first principle for learning in neural circuits, in particular in sensory systems where limited bandwidth requires

efficient coding [43, 54]. Receptive fields learned by RBMs are similar to those found in the brain and are learned by related models such as sparse coding and independent component analysis [30]. Recent work also shows that combinations of Hebbian learning with biologically plausible recurrent network connectivity approximate the contrastive learning in RBMs [55, 56]. We therefore view a trained RBM model as a baseline model of an optimal noisy, spiking information channel.

## The role of ongoing learning and homeostasis

In biology, pruning and learning occur simultaneously. To emulate this, we retrained networks for ten epochs after each batch of pruning. This batched form of sensory-driven plasticity allowed the networks to recover from a performance drop observed immediately after pruning. In biology, numerous types of homeostatic plasticity could compensate for the loss of cells and synapses without error-driven retraining [57]. Synaptic depression and spike-frequency adaptation are examples for such pre- and postsynaptic adaptation mechanisms, respectively [58, 59]. Weight re-scaling resembling such adaptation mechanisms could potentially help make pruning less disruptive. We conjecture that such homeostatic rules may accelerate learning by reducing the amount of training needed to compensate for the effects of pruning.

## Pathological pruning, hallucinations, and neurological disorders

Model networks can provide insight into neurological disorders. For example, Reichert et al. [42] use deep Boltzmann machines to model Charles Bonnet syndrome. In their model, homeostasis re-activated internal representations in "blinded" DBMs, creating hallucinations. Here, we explored how networks degrade when pruning is inaccurate or excessive. Unlike in Reichert et al. [42], our networks continued to receive input. However, aberrant pruning led to partial blindness and neuronal loss. On the MNIST dataset, all pruned models (except the Anti-FI network) still produced distinguishable latent encodings, indicating that the basic circuits of perception were intact. However, pruning degraded the quality and diversity of generated digits.

Statistical theories view hallucinations as errors in internal predictive models [60, 61]. Hallucinated stimuli in the absence of input is one form of this. Over-confidence in erroneous internal models, or incorrect predictions of which stimuli are likely, can also lead to mistaken —or hallucinatory— perception. Loss of generative diversity in our simulations indicates that the model's internal priors no longer match the external world. Interpreted in the context of predictive coding, this bias implies that some stimuli are poorly predicted by internal states, and might therefore register as unexpected or surprising.

This suggests a speculative connection to schizophrenia, for which hallucinations and altered cognition are core symptoms. Schizophrenia also involves pathological neural pruning [8, 11–13]. Could pruning-induced degradation of internal predictive models explain some of the symptoms of this disease? The close relation between perception and reconstruction in DBMs and their hierarchical organization make them an interesting model candidate to further investigate hallucinations. Such modeling may provide theoretical insights into neurological disorders that involve aberrant pruning.

Apart from neurological disorders, understanding pruning is important for understanding learning and cognitive flexibility. In all of our experiments, the sensory encoding task was fixed. Overzealous optimization to a fixed distribution could impair flexibility, a phenomenon that might relate to developmental critical periods and neurological disorders [8]. In biology, the brain must balance optimality against the need to maintain cognitive reserves to support ongoing learning and changing environments.

## Summary and outlook

Overall, we have shown that local activity-dependent synaptic pruning can solve the global problem of optimizing a network architecture. In contrast to pruning rules based on synaptic weights, our information-based procedure readily identified redundant neurons and led to more efficient and compact networks. The pruning strategy we outline uses quantities locally available to each synapse, and is biologically plausible. The artificial neural networks explored here are abstract. If analogous processes operate in biology, then a similar pruning procedure could optimize metabolic cost by eliminating redundant neurons. An important future direction is to analyze these pruning rules in conjunction with synapse formation, which has been hypothesized to provide an additional mechanism for functional optimization of network topologies [20].

# Materials and methods

## Datasets

We used two different datasets of visual image patches to train and evaluate our models. For Figs 1 and 2, we randomly selected 90, 000 circular patches of different size from images of the CIFAR-10 dataset [62] to mimic the encoding of visual scenes through receptive fields of retinal ganglion cells. We then converted images to gray-scale and binarized them by thresholding at the median pixel intensity. The procedure is illustrated in Fig 1A.

For Figs 3 and 4, we used the MNIST handwritten digits dataset [46]. Due to memory limitations on our computational hardware, each square $28 \times 28$ images was cropped to $20 \times 20$ pixels by removing four pixels on each side. This resulted in a visible layer size of 400 units (see Fig 3A). The labeled images belonged to one of ten digit categories (0–9) and were divided in a training set of 60, 000 and a held-out test set of 10, 000 images. The gray-scale images were binarized according to the mean pixel intensity in the training and test set, respectively.

## Model definition and training

RBMs are generative stochastic encoder models consisting of $n$ binary units (neurons) that are organized in two layers: a visible layer $\mathbf{v}$ that directly corresponds to a given input vector and a hidden (or latent) layer $\mathbf{h}$ which contains an encoding of the input data. Each visible layer neuron $v_i$ has undirected weighted connections (synapses) to each hidden neuron $h_j$ and vice versa, but neurons within a layer are not connected to each other (see Fig 1A).

The energy function given in Eq 1 assigns energy to each visible layer pattern. It can be rewritten as a joint probability, which describes the system's Boltzmann distribution at temperature $T = 1$:

$$p(\mathbf{v}, \mathbf{h}) = \frac{1}{Z} e^{-E_{\mathbf{v},\mathbf{h}}^{\phi}}, \tag{7}$$

with $Z$ being the partition function that sums over all possible configurations of visible and hidden neurons. By marginalizing and taking the logarithm, we get:

$$\log p(\mathbf{v}) = \log \left( \sum_{\mathbf{h}} e^{-E_{\mathbf{v},\mathbf{h}}^{\phi}} \right) - \log Z. \tag{8}$$

The training objective is to adjust the parameters $\phi$ such that the energy for a training pattern is lowered compared to the energies of competing patterns [63]. Lowering its energy translates to assigning a greater probability to that pattern. We maximize $p(\mathbf{v})$ by maximizing the first term in Eq 8, i.e. the unnormalized log probability assigned to the training pattern $\mathbf{v}$, and minimizing the second term. From the restricted connectivity it follows that the

probability of a unit being in the on-state is conditionally independent from the states of units of the same layer, given a configuration of the other layer:

$$p(h_j|\mathbf{v}) = \sigma(b_j^h + \textstyle\sum_i v_i w_{ij}), \qquad p(v_i|\mathbf{h}) = \sigma(b_i^v + \textstyle\sum_j h_j w_{ij}). \tag{9}$$

The positive-negative or wake-sleep algorithm [43] exploits this conditional independence and is used to train RBMs. While the positive or wake phase corresponds to a forward-pass through the network ($p(h_j|\mathbf{v})$), the model metaphorically dreams about $p(v_i|\mathbf{h})$ in the negative or sleep phase. Specifically, the visible layer is initialized to a random state and the layers are updated iteratively following alternating Gibbs sampling for $k \to \infty$ times. The goal is for the final sample to originate from the equilibrium distribution of the model. $k$-step contrastive divergence [32] accelerates the learning algorithm by effectively replacing the desired sample from the model distribution by a single reconstruction after $k$ Gibbs steps. The most extreme case is one-step contrastive divergence, where the Gibbs chain is truncated after just one iteration of initializing the visible layer to a training example, sampling the hidden layer, and re-sampling the visible layer.

A deep Boltzmann machine consists of a visible layer and multiple hidden layers. Analogous to Eq 1, the energy function of a two-layer network is given by:

$$E_{\mathbf{v},\mathbf{h}}^{\phi} = -\sum_{i=1}^{n_v} b_i^v v_i - \sum_{j=1}^{n_{h1}} b_j^{h^1} h_j^1 - \sum_{k=1}^{n_{h2}} b_k^{h^2} h_k^2 - \sum_{i=1}^{n_v}\sum_{j=1}^{n_{h1}} v_i h_j^1 w_{ij}^{h^1} - \sum_{j=1}^{n_{h1}}\sum_{k=1}^{n_{h2}} h_j^1 h_k^2 w_{jk}^{h^2}. \tag{10}$$

Our parameter set $\phi$ is thus augmented by an additional weight matrix $\mathbf{W^{h^2}}$ and another bias vector $\mathbf{b^{h^2}}$. Learning in DBMs also follows the positive-negative algorithm, yet layer-wise: each layer of hidden units aims to find an appropriate representation of the distribution over the variables in its preceding layer in order to generate it. However, since contrastive divergence is too slow for training deeper networks [51], a mean-field variational inference approach is used to train DBMs [64].

## Model fitting and sampling

All models were implemented in TensorFlow [65] making use of open-source code for the implementation of RBMs and DBMs (https://github.com/yell/boltzmann-machines). Our pruning functionality was added in the forked repository (https://github.com/carolinscholl/PruningBMs). Gibbs sampling ran on NVIDIA GeForce GTX 980 or 1080 GPUs. For the layer states to be uncorrelated, a sample was stored after every 200th Gibbs step. The number of samples per layer was the same as the number of training instances.

**RBMs for CIFAR-10 patches.** Single-layer RBMs were fit to CIFAR-10 patches using one-step contrastive divergence [32]. The radius of a circle determined the number of pixels and visible units (see Fig 1A). Weights were initialized randomly from a Gaussian distribution with $\mathcal{N}(\mu = 0, \sigma^2 = 0.01)$. The hidden units outnumbered the visible units because we aimed for a sparse representation and uncorrelated hidden units. We also initialized the hidden biases to −2 to encourage sparse activity in the hidden layer. Other than that, no sparsity target was defined. As recommended by [63], visible biases were initialized with $\log[p(v_i)/(1 - p(v_i))]$, where $p(v_i)$ corresponds to the fraction of training instances where pixel $i$ was in the on-state. No mini-batches were used, updates were applied after each training instance. RBMs were trained for 2 epochs, with a decaying learning rate from 0.1 to 0.01 and momentum of 0.9. Weights were not regularized to encourage a sloppy pattern in parameter sensitivities as is characteristic for biological networks [66, 67]. Fitting was repeated 10 times with different random seeds for weight initialization.

**DBMs for MNIST digits.** The comparably large number of 400 pixels of each cropped MNIST image required their segmentation in receptive fields (see Fig 3A). Hidden layer $\mathbf{h}^1$ had the same number of units as the visible layer $\mathbf{v}$ (400). Each unit from $\mathbf{h}^1$ was connected to a rectangle spanning a maximum number of $5 \times 5$ neighboring units from $\mathbf{v}$. A stride of (1, 1) without zero-padding was used, so the receptive fields overlapped and were smaller at the borders of the image. The use of receptive fields reduced the number of weights connecting $\mathbf{v}$ and $\mathbf{h}^1$ from originally $400 \times 400 = 160, 000$ to 8, 836. The fully connected hidden layer $\mathbf{h}^2$ with 676 units then combined the receptive field encodings of parts of the image into a latent representation of the full image.

The deep Boltzmann machine was built from two individual RBMs that were pre-trained for 20 epochs with one-step contrastive divergence. After fitting the first RBM with receptive fields, its hidden units were sampled with $\mathbf{v}$ clamped to one input vector at a time. The resulting binary hidden layer vectors served as training data for the second fully connected RBM. Weights for both RBMs were initialized randomly from a Gaussian distribution with $\mathcal{N}(\mu = 0, \sigma^2 = 0.01)$. Ten different seeds were used for repeated simulations. Visible biases were initialized with $-1$, hidden biases with $-2$. Neither sparsity targets nor costs were defined. Momentum was set to 0.5 for the first 5 epochs and then increased to 0.9 as recommended by [63]. The learning rates decreased on a logarithmic scale between epochs, starting from 0.01 for the first RBM, and from 0.1 for the second RBM, to 0.0001 in their final epochs of training. No mini-batches or weight regularization methods were used.

When stacking the two RBMs, the hidden layer of the first RBM and the visible layer of the second RBM were merged by averaging their biases. The resulting deep Boltzmann machine with two hidden layers was trained jointly for 20 epochs following a mean-field variational inference approach using persistent contrastive divergence with 100 persistent Markov chains [64]. A maximum value of 6 was defined for the norm of the weight matrix to prevent extreme values. The hidden unit firing probabilities decayed at a rate of 0.8. Neither a sparsity target nor weight regularization were applied and parameters were updated immediately after presenting a training image.

## Pruning criteria

We compared six different weight pruning criteria throughout our simulations, three of which targeted the removal of weights that carried low FI. For small RBMs, computing the full FIM and its eigenvectors was feasible, using code from github.com/martinosorb/rbm_utils. The parameter-specific entries of the first eigenvector were then used as indicators of parameter importance. For DBMs we only used two of the three FI pruning criteria: Variance FI refers to estimating the weight-specific FIM diagonal entries by computing (4) based on samples from all layers of the models. Heuristic FI does not directly track the firing rates, but uses the mean rates instead (see (6) and Materials and Methods: *Heuristic estimate of Fisher Information*). For the deep Boltzmann machine, the FIM diagonal was computed layer-wise assuming weakly correlated hidden units. To examine the relevance of high FI parameters, we also deleted the *most* important weights according to the variance estimate of FI, which we refer to as Anti-FI pruning. When we used the weight magnitude as a proxy of importance, we deleted weights with lowest absolute value. As a baseline, we also removed a randomly chosen sample of weights. As an additional control case, we randomly removed whole units, thereby pruning a comparable number of weights as the other criteria.

**Pruning procedure.** To simulate synaptic pruning, weight importance was estimated according to the selected pruning criterion. According to this criterion, a threshold was set at a pre-specified percentile. All weights whose estimated importance did not meet this threshold

were removed from the network by fixing their values to zero. For RBMs trained on CIFAR-10 patches, the threshold corresponded to the 50th percentile of weight importance, meaning that half of the weights were removed. For DBMs trained on MNIST, the threshold corresponded to the 10th percentile of weight importance for $\mathbf{h^1}$ and to the 25th percentile for $\mathbf{h^2}$. If all incoming weights to a hidden unit were pruned, the unit was removed from the network. In DBMs, a hidden unit was also deleted if all its outgoing weights were pruned. After pruning, the model was re-initialized with unaltered values of the remaining parameters for the next pruning iteration. The RBMs fit to CIFAR-10 patches were retrained for 2 epochs after each of 3 pruning iterations. For DBMs, the retraining period was shortened from 20 to 10 epochs and mini-batches of size 10 were used to accelerate retraining after each pruning iteration. DBMs were pruned for a maximum of 10 iterations. In Figs 3 and 4 we present the results after a comparable total number of remaining weights. The results after completion of 10 iterations for all criteria can be found in Figs A and B in S2 Text.

## Evaluation

Each experiment started by fitting a sufficiently large model to the data. While the visible layer size was determined by the number of pixels in each training image, the (final) hidden layer was set to be larger. The resulting over-parameterized model was iteratively pruned, while its fit was monitored.

For small RBMs that were trained on CIFAR-10 patches, we evaluated a model's generative performance by comparing the probabilities of generated visible layer samples to the probabilities of patterns occurring in the training data. Furthermore, we computed the Kullback-Leibler divergence between the data distribution and the distribution of generated samples $D_{KL}(\text{data} \,||\, \text{model})$.

For DBMs that were trained on MNIST digits, we made use of the labeled data to evaluate both the encoding and generative performance.

First, we considered the encodings of the data in the final hidden layer of the network. While the visible layer was clamped to one training instance at a time, the hidden unit activations were sampled. We expect these latent representations to comprise a more useful training set for the classifier than the raw images. The resulting set of 60,000 final hidden layer encodings was used to train a multinomial logistic regression classifier, which had to distinguish between the 10 digit categories. We refer to the classification accuracy of this classifier built on top of the final hidden layer as the encoding quality of the model.

Second, we evaluated the patterns generated by the network. Since Boltzmann machines try to approximate the data distribution with their model distribution, these generated patterns should ideally resemble digits. Thus, we trained a multinomial logistic regression classifier on the 60,000 raw MNIST images. After training, this classifier received patterns generated by the network. For each of the ten digit classes, it returned a probability of the current sample belonging into it. The argmax over this probability distribution was used to assign the class. The average of the winning class probabilities was used as a confidence score. It served as a measure of digit quality. Furthermore, the entropy of the distribution of assigned classes served as a measure of digit diversity. A maximum entropy of approximately 2.30 nats corresponds to completely balanced digits.

Moreover, the generative performance was compared to that of a classifier trained on the raw digits. The quality of generated digits was compared to the quality of the 10,000 held-out MNIST test digits and to the quality of random patterns, using the same classifier.

## Local estimates of Fisher Information

### Variance estimate of Fisher information

Parameter importance is reflected in the curvature of the energy landscape of an RBM when slightly changing two parameters. Computing this for each parameter pair leads to the FIM (see Eq 2), where $F_{ij}$ stands for the Fisher information of the considered couple ($\phi_i$, $\phi_j$). The entries of the FIM thus have the form [30]:

$$
\begin{aligned}
F_{w_{ij}, \, w_{kl}} &= \langle v_i h_j v_k h_l \rangle - \langle v_i h_j \rangle \langle v_k h_l \rangle, & F_{w_{ij}, \, b_k^v} &= \langle v_i h_j v_k \rangle - \langle v_i h_j \rangle \langle v_k \rangle, \\
F_{w_{ij}, \, b_k^h} &= \langle v_i h_j h_k \rangle - \langle v_i h_j \rangle \langle h_k \rangle, & F_{b_i^v, \, b_j^h} &= \langle v_i h_j \rangle - \langle v_i \rangle \langle h_j \rangle, \\
F_{b_i^v, \, b_j^v} &= \langle v_i v_j \rangle - \langle v_i \rangle \langle v_j \rangle, & F_{b_i^h, \, b_j^h} &= \langle h_i h_j \rangle - \langle h_i \rangle \langle h_j \rangle.
\end{aligned}
$$

The diagonal of the FIM corresponds to evaluating changes in the energy landscape of the model when perturbing just one parameter. The importance of a weight thus simplifies to the average coincident firing of pre- and postsynaptic neurons. The importance of a bias value is estimated by the variance of a neuron's firing.

$$
F_{w_{ij}, w_{ij}} = \langle v_i^2 h_j^2 \rangle - \langle v_i h_j \rangle^2, \quad F_{b_i^v, b_i^v} = \sigma_{v_i}^2, \quad F_{b_j^h, b_j^h} = \sigma_{h_j}^2.
$$

We consider this estimate to be locally available statistics to the individual neuron.

### Heuristic estimate of Fisher information

In this section, we derive a mean-field approximation estimate of $\langle v_i h_j \rangle$ in terms of a small deviation from the case where presynpatic and postsynaptic activity are statistically independent, $\langle v_i h_j \rangle \approx \langle v_i \rangle \langle h_j \rangle$ that accounts for correlated activity introduced by the synapse between neurons $i$ and $j$. Starting from Eq 4, we get

$$
F_{w_{ij}, w_{ij}} = \langle v_i h_j \rangle (1 - \langle v_i h_j \rangle).
$$

This can be computed if there is a local mechanism for tracking and storing the correlation of pre- and postsynaptic firing rates $\langle v_i h_j \rangle$. This expectation is closely related to more readily available local statistic, like the mean rates and weight magnitude. Since $v_i$ and $h_j$ are binary in $\{0, 1\}$, the expectation $\langle v_i h_j \rangle$ amounts to estimating the probability that $v_i$ and $h_j$ are simultaneously 1, i.e. $p(v_i = 1, h_j = 1)$. One can express this in terms of a mean rate and the conditional activation of presynaptic neuron given a postsynaptic spike, using the chain rule of conditional probability:

$$
p(v_i = 1, h_j = 1) = p(v_i = 1 | h_j = 1) \cdot p(h_j = 1) = \langle h_j \rangle p(v_i = 1 | h_j = 1).
$$

If the hidden layer size is large and the hidden units are independent, we may approximate the activity of all *other* hidden units apart from a particular $h_j$ using mean-field. The activation of a visible unit is given by Eq 9. We replace the hidden units by their mean firing rate:

$$
p(v_i = 1 | \langle \mathbf{h} \rangle) = \sigma(b_i^v + \mathbf{W}_i \langle \mathbf{h} \rangle).
$$

To estimate the contribution of all units *except* $h_j$, one computes:

$$
p(v_i = 1 | \langle \mathbf{h} \backslash h_j \rangle) = \sigma(b_i^v + \mathbf{W}_i \langle \mathbf{h} \rangle - w_{ij} \langle h_j \rangle).
$$

To get the mean-field activation assuming $h_j = 1$,

$$
\begin{aligned}
p(v_i = 1 | h_j = 1) \quad &\approx \Pr(v = 1 | \langle \mathbf{h} \backslash h_j \rangle, h_j = 1) \\
&= \sigma(b_i^v + \mathbf{W}_i \langle \mathbf{h} \rangle - w_{ij}\langle h_j \rangle + w_{ij} \cdot 1) \\
&= \sigma(b_i^v + \mathbf{W}_i \langle \mathbf{h} \rangle + w_{ij}(1 - \langle h_j \rangle)).
\end{aligned}
$$

We can obtain an alternative (and, empirically: more accurate) mean-field approximation using the average firing rate of the visible unit, $\langle v_i \rangle$. Given this mean rate, we can estimate the activation as $\sigma^{-1}(\langle v_i \rangle)$. This estimate can replace the $b_i^v + \mathbf{W}_i \langle \mathbf{h} \rangle$ terms, leading to:

$$
\begin{aligned}
p(v_i = 1 | h_j = 1) \quad &\approx \sigma(\sigma^{-1}[\langle v_i \rangle] + w_{ij}[1 - \langle h_j \rangle]) \\
&= \left\{ 1 + \exp\left[ -\ln \frac{\langle v_i \rangle}{1 - \langle v_i \rangle} - w_{ij}[1 - \langle h_j \rangle] \right] \right\}^{-1} \\
&= \left\{ 1 + \frac{1 - \langle v_i \rangle}{\langle v_i \rangle} e^{-w_{ij}[1 - \langle h_j \rangle]} \right\}^{-1}.
\end{aligned}
$$

This implies

$$
\langle v_i h_j \rangle \approx \langle v_i \rangle \langle h_j \rangle \{ \langle v_i \rangle + [1 - \langle v_i \rangle] e^{-w_{ij}[1 - \langle h_j \rangle]} \}^{-1}.
$$

This has an intuitive interpretation: for fixed $\langle v_i \rangle$ and $\langle h_j \rangle$, the coincidence in presynaptic and postsynaptic firing is a sigmoidal function of the weights. This implies the magnitude of synaptic weights, when combined with information about average firing rates, provides a useful proxy for computing pre-post correlations, and therefore estimating a synapse's importance in the network.

## Supporting information

**S1 Text. Latent activity may indicate optimal model size.**
(PDF)

**S2 Text. Comparison after 10 pruning iterations.**
(PDF)

## Author Contributions

**Conceptualization:** Carolin Scholl, Michael E. Rule, Matthias H. Hennig.

**Formal analysis:** Carolin Scholl, Michael E. Rule.

**Funding acquisition:** Matthias H. Hennig.

**Investigation:** Carolin Scholl, Matthias H. Hennig.

**Methodology:** Carolin Scholl, Michael E. Rule, Matthias H. Hennig.

**Software:** Carolin Scholl.

**Supervision:** Michael E. Rule, Matthias H. Hennig.

**Validation:** Carolin Scholl, Michael E. Rule, Matthias H. Hennig.

**Visualization:** Carolin Scholl, Michael E. Rule.

**Writing – original draft:** Carolin Scholl, Michael E. Rule, Matthias H. Hennig.

**Writing – review & editing:** Carolin Scholl, Michael E. Rule, Matthias H. Hennig.

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
