## [Decision Letter · Decision Letter 0]

13 Apr 2021

Dear Dr. Hennig,

Thank you very much for submitting your manuscript "The Information Theory of Developmental Pruning: Optimizing Global Network Architecture Using Local Synaptic Rules" for consideration at PLOS Computational Biology.

As with all papers reviewed by the journal, your manuscript was reviewed by members of the editorial board and by several independent reviewers. In light of the reviews (below this email), we would like to invite the resubmission of a significantly-revised version that takes into account the reviewers' comments. With regards to the question of relevance for biology, we encourage you to highlight the important algorithmic-level links between the type of systems you are studying here and real neural circuits.

We cannot make any decision about publication until we have seen the revised manuscript and your response to the reviewers' comments. Your revised manuscript is also likely to be sent to reviewers for further evaluation.

Sincerely,

Blake A Richards

Associate Editor

PLOS Computational Biology

Lyle Graham

Deputy Editor

PLOS Computational Biology

Reviewer's Responses to Questions

**Comments to the Authors:**

Reviewer #1: The paper investigates the pruning of connections with low Fisher Information in restricted and deep Boltzmann machines. This approach is compared with pruning based on a locally computable proxy of Fisher information, the maximum Fisher information, and the synaptic weight, as well as random pruning.

As the most interesting property, the authors demonstrate that the low FI approach typically removes all connections to a hidden neuron, such that the synapse pruning can also implement neuron pruning. At the same time, the pruned networks retain high encoding/classification and generative performance. The methods the authors use to demonstrate their finding are sound and appropriate. However, I am missing error estimates or a validation by multiple trials.

Moreover, although the topic of pruning is a very interesting and relevant topic both in biological and artificial neural networks, I think the findings and writing of the paper are more on the artificial intelligence than on the biological networks side (usage of RBM, training on MNIST/CIFAR, no comparison to experimental data), which should be changed to be suitable for publication in this journal.

Major points:

Errors:

It seems as if all the results stem from single network instances and there are no error estimates reported. As the pruning and RBM training heavily depend on the initial network, I think it would be good to compare results over multiple initialization.

Link to biology:

The bidirectional weights and their training within RBMs are not exactly biological. Thus, it is unclear whether the advantage FI provides over weight-dependent pruning will still hold in biologically more realistic settings. Moreover, if the biological realistic implementation of an information based learning rule is in the focus, it should be highlighted and discussed better in the results instead of being hidden in the methods/appendix.

Neuron removal:

Throughout the paper, the authors often stress the fact that FI-based approaches prune connections such that whole neurons can be removed. I feel like there is a proper control missing for this to demonstrate the superiority of the FI approach. I would propose to introduce a pruning strategy which removes whole neurons (either randomly or by minimum sum of incoming weights). In that way, there would be a control case with similar neuron number to compare with.

Minor points:

all Figures: Colors of Random and weight-dependent pruning were not so well discernible for me.

p.2 l.81: it should be b^h in the parameter-set instead of h^h. Also, I would move the sentence about the Bernoulli RBM after the description of the activities, which are the binary quantities

p.3 l.87 I would move the discussion of diagonality after the example FI (Eqs.2/3), because i think these examples would make it easier to understand that non-local information is needed for the full FI

p.3 l.95 I am not completely sure what is meant by redundant here. Is each parameter redundant for the network or are they redundant (covariant or so) w.r.t. each other? Pleas clarify.

Fig 1C/D: the labels h and v were a bit confusing. Shouldn't it be b^v and b^h?

Fig 2: It is hard to see a strong effect in the pattern distributions. Is there a way to make this clearer? Possibly show ratios?

Also the demonstrated changes in distributions will strongly depend on the size of the model and initial overparametrization

p.7 l 210 The explanation on generative performance is a bit distracting here. I would suggest moving it to the respective section.

p.7 l 223 The reason for calculating the percentile is unclear. Are these the pruned synapses? Please clarify! Also, making more clear that these are the 10% of the weights with the lowest FI, would make sense.

p.7 l.227 The reading flow here was a bit unsteady. Possibly move the statement of the before-pruning accuracy to line 210, where they are introduced.

p.7 l. 236 Isn't this a consequence of your FI-estimator being zero for many more units than the weight such that you remove more connections. I think the plot 3b is a fair way to compare the pruning strategies. If you want to compare, let the weight-pruning run longer.

Fig. 3C: Plotting over n_wh does not add much here. I would plot over the iteration number for comparability.

Fig. 4D: Why invert the x-axis?

p.9 l.311 *Anti-FI

p.9 l.300 Typo "versus"

l. 353 There is a lot of experimental and theoretical work on connectivity overshoot, e.g. by Arjen van Oojen and Jaap van Pelt (based on activity dependent rewiring). Also spine turnover (especially removal) has been demonstrated to be much higher during maturation. Would this be the direction this is aiming at? What would be the functional use of a "critical phase"?

Reviewer #2: A salient feature of neural development is pruning: the number of neurons and connections first grows, but at some point both decrease. It is not known why this happens, or which connections and neurons are pruned. The authors address the latter question, and propose that weights that have the smallest effects on activity, as measured by the Fisher information, are preferentially pruned; and if all connections from or to a neuron are pruned, the neuron is pruned as well.

I'm not an expert, but this is, I believe, a novel and interesting hypothesis, and will make a nice contribution to the field. My comments are almost exclusively about presentation -- there were a lot of places I simply got lost. That didn't really detract from the big picture, but it would be nice if things were clarified.

1. Sloppiness is mentioned in the abstract, and not again until methods. It would make sense to me to either drop it in the abstract or mention it in the main text.

2. I personally would drop initials, and spell out RBM, DBM, FIM and FI. Or at least spell out the first three. I'm not sure why anybody uses initials; in my view authors should be allowed to use at most one. There's actually a reason for that: people rarely read papers beginning to end, and it's very annoying to have to hunt through a paper to look for the meaning of initials.

3. Given the definition of the energy, I believe Eq. 2 is wrong: the derivative should be with respect to the partition function, not the energy. If so, that should be corrected.

4. I don't think P_v,h was ever defined.

5. l 95-6: "When F_ij tends towards zero, the two parameters phi_i and phi_j are redundant." I can't for the life of me even guess what that means. It should be explained. Or dropped; I don't think it was ever used.

6. l 99-100: "The resulting entries of the FIM depend on coincident firing of pre- and postsynaptic neurons and are arguably locally available." How can something that depends on two presynaptic and two postsynaptic neurons be locally available?

7. I could not figure out what's in fig. 1c and d. This should be explained much more clearly.

8. l 115-7: "The correspondence between parameter importance estimated from the first eigenvector and from the diagonal supports our use of Optimal Brain Damage for larger models, when computing the full FIM was no longer feasible." Couldn't make sense of this. It doesn't help that Optimal Brain Damage was (I believe) never explained.

9. l 117-22:. "Strikingly, the important weights typically aligned with few hidden units and their biases. This structure of the FIM suggests a separation into important hidden units and unimportant ones. It follows that FI motivated pruning likely leads to entire units becoming disconnected, which would allow their removal from the network. This would correspond to neuron apoptosis after excessive synaptic pruning." The second two sentences make sense. But the first two don't, and so it's not clear how the second follow from the first.

10. l 131-3: "For a pruning criterion based on the full FIM, we used the weight-specific entries of its first eigenvector as a direct indicator of weight importance." Couldn't make sense of this.

11. After Eq. 4, I think it's important to write down the expression for Fisher information (Eq. 19) that was actually used, written in a human-readable form. As far as I can tell, Eq. 19 can be written

<v_i h_j=""> = <v_i><h_j>/[<v_i> + (1-<v_i>)exp(-w_ij (1-<h_j>))].

Algebra mistakes are possible, but I believe the correct expression looks something like this. And it's kind of easy to make sense of: besides the dependence on v_i and h_j, it's an approximately sigmoidal function of the weights. So it would be nice to include it.

12. l 160: "Generally, excessive pruning was detrimental to generative performance" It seems that it's only for the orange line (Heuristic FI?) that pruning was detrimental to generative performance.

13. Fig. 2 caption: "For all pruning strategies except Anti-FI pruning the model retains the ability to match the distribution of the training data (dashed lines) after retraining, indicating good generative performance." As far as I can tell, Anti-FI matched the true distribution in all but one panel.

14. l 186-8: "In sum, for all pruning strategies (except Anti-FI pruning, which removed a large fraction of important weights), the network could recover from the loss of weights and units through retraining". It looks to me like Anti-FI recovers as well.

15. mnist is 28x28. why scale to 20x20? The reason for this should probably be explained.

16. Fig. 3B: why did random and [w] stop at 10^5?

17. fig. 4: probability may not be the best measure -- it's possible in principle that probability deteriorates but performance stays high. I think it would be good to report both performance and probability.

And again, why stop at 10^5 for random and [w], and even higher for Anti-FI?

18. l 350-3: "A recent study of the effects of perturbing the input during different time points of training in neural networks suggests that a critical learning period may be visible in a plateau of the FIM trace [45]." I couldn't make sense of this.</h_j></v_i></v_i></h_j></v_i></v_i>

**Have all data underlying the figures and results presented in the manuscript been provided?**

Reviewer #1: Yes

PLOS authors have the option to publish the peer review history of their article (what does this mean?). If published, this will include your full peer review and any attached files.

Reviewer #1: No

Reviewer #2: No

**Have the authors made all data and (if applicable) computational code underlying the findings in their manuscript fully available?**

Reviewer #2: Yes
---

## [Decision Letter · Decision Letter 1]

19 Aug 2021

Dear Dr. Hennig,

Thank you very much for submitting your manuscript "The Information Theory of Developmental Pruning: Optimizing Global Network Architecture Using Local Synaptic Rules" for consideration at PLOS Computational Biology. As with all papers reviewed by the journal, your manuscript was reviewed by members of the editorial board and by several independent reviewers. The reviewers appreciated the attention to an important topic. Based on the reviews, we are likely to accept this manuscript for publication, providing that you modify the manuscript according to the review recommendations.

Please be sure to address the remaining smaller comments from the reviewers. If you feel that the additional analyses proposed by reviewer 1 are outside the scope of this particular paper (which the reviewer recognised they may be), please just state this clearly in your response to reviewers and consider adding commentary in the discussion in this direction.

Sincerely,

Blake A Richards

Associate Editor

PLOS Computational Biology

Lyle Graham

Deputy Editor

PLOS Computational Biology

[LINK]

Please be sure to address the remaining smaller comments from the reviewers. If you feel that the additional analyses proposed by reviewer 1 are outside the scope of this particular paper (which the reviewer recognised they may be), please just clarify and consider adding commentary in the discussion in this direction.

Reviewer's Responses to Questions

**Comments to the Authors:**

Reviewer #1: The authors put great effort into improving the paper and addressed most of my concerns. I only have a few remaining issues (and suggestions) before I recommend publication:

Concerning 1.4: Random unit pruning:

(a) "... in the one-layer RBM all visible units are connected to all hidden units.When a hidden unit is removed here, the activity and Fisher information can completely re-arrange with re-training". As the unit pruning seems to be the main advantage of the FI-approach, I think this control case should also be included for the single layer RBM. The above intuition can the be discussed and demonstrated in Fig 2C: One would see a large divergence before but not after retraining.

(b) From what I see in Figure 3, the random unit pruning is not really fair with respect to the units in hidden layer 1, as it removes an order of mangitude more neurons in that layer. To really demonstrate that FI-pruning leads to a better network structure more quickly, I propose to adapt this and remove less neurons in h1 to arrive at a comparable structure

l.134/l.333f Could you provide more motivation why it is easier to track firing rates and weights instead of correlation and weights. Biologically, Ca or CaMKII are thought to be local proxies of correlated activity, but I am not aware of molecular signals tracking especially the presynaptic rate.

l.165 I guess here you need to discuss the results a bit deeper, as otherwise panel 2C would have been sufficient to make the point. Specifically, I noticed that the generative performance only seems to be poor for seldom patterns whereas the performance for abundant patterns seem to match (although with larger variation in the Anti-FI case). Is this really so bad for neural system? From an information theoretic viewpoint, they are surely the most informative patterns. However, as these unmatched patterns are rare, the error introduced by them may be negligible.

l.372 The statement seems a bit bold. Maybe use "activity-dependent pruning that aims to identify uninformative neurons"

Suggestions to improve readability:

- In my opinion, it would make sense to move the introduction of the RBMs (l.23-33) to the end of the introduction (after l.55)

- l.70 Maybe one could also mention the relation between energy and pattern probability in equation 1.

- l.101 I would mention how the models were fitted here (wake sleep algorithm).

- l.101 It is not immediately clear what is meant by "parameter-wise" (first mention). I would stick to the terms full and diagonal or at least specify what is meant in this sentence. Moreover, I think it is may be less confusing to discuss the results in the order they are presented in the figure and move the Also, an activity dependent form is only available from Equation 3 or 4, right?

- l.141 It is not immediately clear why the FI introduced before is "variance" based. Maybe the term could be introduced together with the method and the motivation of "variance" could be explained.

- l.150 I think it should be shortly motivated what the generative performance means/relates to in the neuronal/biological system, to give a better intuition what the FI-approach actually preserves.

Finally, I would have another suggestion:

Another advantage of the FI-dependent pruning over other methods may be the fact that it could be used to determine when pruning should be stopped. At the moment this is not the case as the lowest-FI quantile of synapses is always removed. If, instead, only synapses below an FI-threshold would be removed, pruning would naturally stop if all synapses have high FI. Such a convergence would remove the necessity to select a suitable number of pruning iterations for the model and prevent the performance loss of the FI-based models after massive pruning in Fig 3. Assuming that pruning stops after all synapses have high FI, one would get one "optimal" pruned model (instead of one per pruning iteration). Determining these optimal models for different input statistic would also allow predictions on the number of surviving synapses and neurons as well as weight distributions (for example comparing the networks after training with a 5-class MNIST subset and the full dataset). Varying the input statistics and getting different resulting models would greatly underline the point that FI-pruning actually selects input-related "optimal" model architectures and not just "smaller" models whose size is determined by the number of iterations. Moreover, such an analysis would provide more insight into the relation between the encoding of the Boltzmann machine and optimal pruned models, which, I guess, was a goal of this line of research.

The differences in the resulting optimal networks could, in turn, be compared with existing data on network complexity/ neuron and spine densities in animals reared in different environments (e.g. dark rearing, rearing with differently oriented bars, normal cages, enriched environments). This would make a nice connection to biology and provide actually testable pre/postdictions (Concerning the experiments you proposed: at least the experimentalists I know say that it is not feasible to track pre- and postsynaptic activity and the weight of an identified synapse over time at the moment).

I am aware that this additional analysis may be work-intensive and beyond the scope of this paper. However, I think it may greatly improve the manuscript or at least provide an interesting direction for future research.

Reviewer #2: The paper is _much_ improved, and I'm happy with it. Only two comments:

1. in Eq. 10, I believe the weights should have superscripts.

2. I would suggest moving A1 and A2 to Methods. I suspected the more mathematically inclined will be interested. I certainly was, since I got it wrong the first time around. ;) This is, though, completely up to the authors.

**Have the authors made all data and (if applicable) computational code underlying the findings in their manuscript fully available?**

Reviewer #1: Yes

Reviewer #2: Yes

PLOS authors have the option to publish the peer review history of their article (what does this mean?). If published, this will include your full peer review and any attached files.

Reviewer #1: No

Reviewer #2: No

Figure Files:

Data Requirements:

Reproducibility:

References:

---

## [Editor Report · Decision Letter 2]

17 Sep 2021

Dear Dr. Hennig,

We are pleased to inform you that your manuscript 'The Information Theory of Developmental Pruning: Optimizing Global Network Architecture Using Local Synaptic Rules' has been provisionally accepted for publication in PLOS Computational Biology.

Best regards,

Blake A Richards

Associate Editor

PLOS Computational Biology

Lyle Graham

Deputy Editor

PLOS Computational Biology

---

## [Editor Report · Acceptance letter]

6 Oct 2021

PCOMPBIOL-D-20-02149R2 

 The Information Theory of Developmental Pruning: Optimizing Global Network Architectures Using Local Synaptic Rules

Dear Dr Hennig,

I am pleased to inform you that your manuscript has been formally accepted for publication in PLOS Computational Biology. Your manuscript is now with our production department and you will be notified of the publication date in due course.

With kind regards,

Andrea Szabo
